# Impact of Glass Composition on Hydrolytic Degradation of Polylactide/Bioactive Glass Composites

**DOI:** 10.3390/ma14030667

**Published:** 2021-02-01

**Authors:** Inari Lyyra, Katri Leino, Terttu Hukka, Markus Hannula, Minna Kellomäki, Jonathan Massera

**Affiliations:** 1Faculty of Medicine and Health Technology, Tampere University, 33720 Tampere, Finland; inari.lyyra@tuni.fi (I.L.); katri.j.leino@gmail.com (K.L.); markus.hannula@tuni.fi (M.H.); minna.kellomaki@tuni.fi (M.K.); 2Faculty of Engineering and Natural Sciences, Chemistry and Advanced Materials, P.O. Box 541, Tampere University, 33720 Tampere, Finland; terttu.hukka@tuni.fi

**Keywords:** polylactide, bioactive glass, composites, hydrolytic degradation, thermal degradation, in vitro, mechanical properties, mechanical strength, ion release, molar mass

## Abstract

Understanding the degradation of a composite material is crucial for tailoring its properties based on the foreseen application. In this study, poly-L,DL-lactide 70/30 (PLA70) was compounded with silicate or phosphate bioactive glass (Si-BaG and P-BaG, respectively). The composite processing was carried out without excessive thermal degradation of the polymer and resulted in porous composites with lower mechanical properties than PLA70. The loss in mechanical properties was associated with glass content rather than the glass composition. The degradation of the composites was studied for 40 weeks in Tris buffer solution Adding Si-BaG to PLA70 accelerated the polymer degradation in vitro more than adding P-BaG, despite the higher reactivity of the P-BaG. All the composites exhibited a decrease in mechanical properties and increased hydrophilicity during hydrolysis compared to the PLA70. Both glasses dissolved through the polymer matrix with a linear, predictable release rate of ions. Most of the P-BaG had dissolved before 20 weeks in vitro, while there was still Si-BaG left after 40 weeks. This study introduces new polymer/bioactive glass composites with tailorable mechanical properties and ion release for bone regeneration and fixation applications.

## 1. Introduction

Biodegradable polymers have been in clinical use in orthopedics for many decades. They are advantageous compared to the conventional implants because of their degradability, omitting the need for a removal operation. Polylactides (PLA) is a family of the most well-known biodegradable polymers and has been widely used as pins, plates, and screws [1,2,3,4]. Their advantages include tailorable biodegradation rate, good biocompatibility, and relatively good processability. However, most polymers are bioinert materials, regardless of whether they are degradable or stabile, and bone does not bond on their surfaces [3]. This drawback has been compensated by adding bioactive bioceramics to PLA. Most commercially available biocomposites are based on PLA and calcium phosphate bioceramics (tricalcium phosphate or hydroxyapatite). While these ceramics are the most widely used in clinics, they dissolve slowly and are more likely osteoconductive than osteoinductive [5,6]. Bioactive glass (BaG) is a promising group of bioceramics since they are known to be both osteoconductive and -inductive, depending on their composition [7]. Especially silicate-based BaGs (Si-BaG) have found their way to the clinics in bone reconstruction and regeneration [8,9]. One of the drawbacks of the traditional Si-BaGs is the non-congruent dissolution. The alkaline and alkaline earth ions are released first, leaving a silica-rich surface layer behind [7]. Aside from the typical Si-BaGs, phosphate BaGs (P-BaG) have also gained interest due to their congruent dissolution, enabling a controlled ion release [10,11]. Among the most significant advantages of Si- and P-BaGs are that i) their dissolution rate can be tailored from days to years and ii) any therapeutic ions of interest can be incorporated in their composition [12,13]. However, as all ceramics, BaGs are brittle and hard to shape. Adding BaG, either as fibers or particles, into PLA generates machinable composite materials releasing therapeutic ions enhancing osteogenesis, vascularization, and antimicrobial properties [14]. The local pH change during the degradation of the PLA/Si-BaG composites is also milder compared to either material alone as the basic degradation products of Si-BaGs buffer the acidic degradation products of PLA [15,16].

Controlling the implant material’s degradation rate is crucial to match it to the target tissue’s regeneration rate. The degradation process of PLA is complex, but it is affected by at least the chemical characteristics (crystallinity, chirality, molecular weight), processing conditions (thermal degradation during processing), degradation environment, and size and shape of the sample [17,18,19]. Adding an inorganic phase (for example, BaG) further increases the complexity of the polymer degradation.

In many studies, the degradation rate of PLA in vitro has been found to increase with the addition of BaG. Wang et al. noted that Si-BaG fibers accelerated the degradation of poly-L,DL-lactide (PLDLA) in vivo [20]. However, the composites showed superior bone formation capacity compared to the plain polymer, especially at later time points (26 and 52 weeks). Houaoui et al. reported a faster decrease in the molecular weight of PLDLA in PLDLA/13-93 BaG composites during 10 weeks in Tris compared to PLDLA only [21]. Vergnol et al. reported that adding 45S5 bioactive glass to PLDLA 70/30 increased the mass loss and water absorption of the composite and led to a faster decrease in molecular weight of PLDLA [22]. Rich et al. studied poly(ε-caprolactone-*co-*DL-lactide)/S53P4 glass composites in simulated body fluid for 24 weeks [23]. They observed an increase in water absorption and the degradation rate of the copolymer, too. Felfel et al. reported that the molecular weight of PLA decreased faster in vitro in composites with P-BaG than the plain PLA [24]. However, the opposite results have also been obtained. Navarro et al. did not notice a significant difference in the degradation rate (molecular weight) of PLA in a composite with PLA and P-BaG G5 compared to the plain PLA [25]. For the self-reinforced PLA/BaG composites, both Niemelä et al. and Niiranen et al. reported a more rapid decrease in the molecular weight of the plain self-reinforced PLDLA70 (SRPLA70) compared to the composites of PLA70 and Si-BaG in vitro [15,26]. 

Most of the studies reporting the PLA/BaG composites degradation have used only one BaG and concentrated on the degradation of the polymer. Only a few studies have used P-BaGs or compared different BaGs in composites. In the present study, the main aims were to compare the influence of two different glasses on the degradation behavior of PLA70 in the composites and investigate how the glasses dissolve in the composites. We chose to study two different glasses, a more stable silicate glass (Si-BaG, a composition close to the FDA-approved Si-BaG 13–93) and a more reactive phosphate glass (P-BaG). The BaGs also show different dissolution mechanisms. P-BaG dissolves congruently, and from Si-BaG some ions are released faster than others. The degradation characteristics of the composites were studied over 40 weeks in Tris buffer solution. The results obtained in this work will help further understand the effect of the BaG composition on the composites’ degradation characteristics.

## 2. Materials and Methods

### 2.1. Materials

The biodegradable polymer used in the current study was medical grade poly-L, DL-lactide 70/30 (PLA70; Resomer LR 706 S, Evonik Industries AG, Darmstadt, Germany). The polymer’s inherent viscosity is 4.0 dL/g, and the residual monomer content < 0.05 wt.% (both reported by the manufacturer).

The bioactive glasses used were 1) a silicate-based bioactive glass (Si-BaG) and 2) a phosphate-based bioactive glass (P-BaG). Their compositions are presented in Table 1.

The reagents used for making the glasses were analytical grade Na_2_CO_3_, CaCO_3_, K_2_CO_3_, SrCO_3_, MgO, NH_4_H_2_PO_4_, NaPO_3,_ and Belgian quartz sand. The Ca(PO_3_)_2_ and Sr(PO_3_)_2_ for the production of P-BaG were prepared as in [27]. The reagents for Si-BaG and P-BaG were mixed separately in a platinum crucible. The Si-BaG was melted at 1425 °C for 3 h and the P-BaG at 1100 °C for 30 min. The melt was poured into a graphite mold and annealed at T_g_−15 °C (520 °C and 400 °C for Si-BaG and P-BaG, respectively) for 8 h. The T_g_ of the glasses was obtained by thermal analysis. Finally, the glasses were crushed with a ceramic pestle and mortar by hand and sieved to 125–250 µm particles. The density of bulk glass materials was measured by Archimedes’ principle using deionized water. The accuracy was higher than 0.02 g/cm^3^.

### 2.2. Manufacturing of the Polymer and Composite Rods

Before processing the rods, the PLA70 and the glass powders were dried in a vacuum oven for 8 h at 80 °C to remove excess moisture and air. Cylindrical rods with a diameter of approximately 2 mm were melt-extruded under a nitrogen atmosphere with a custom-made twin-screw extruder Mini ZE 20 × 11.5 D (Neste Oy, Koelaitepalvelut, Porvoo, Finland). Rods containing 1) unfilled PLA70, 2) PLA70 and Si-BaG (target contents 10, 30 and 50 wt.%, denoted as Si-BaG10, Si-BaG30 and Si-BaG50) and 3) PLA70 and P-BaG (target contents 10, 30 and 50 wt.%, denoted as P-BaG10, P-BaG25 and P-BaG35) were manufactured. The composites were denoted according to the actual glass content. See the Results and Discussion Section 3.1 for more details. The processing parameters are gathered in Table 2.

To obtain good quality composite extrudate, there is a need to maintain the pressure at a specific pressure range (which depends, e.g., on the extruder and the die used). If the processing parameters and materials used are influencing the pressure, then for example the processing temperature must be adjusted accordingly. Therefore, in this case, when the BaGs are affecting the viscosity of the polymer, the processing parameters have been changed, resulting in different temperatures and pressures for the different material combinations. 

After processing, the glass content, mechanical properties of the rods, and molecular weights of the PLA70 in the manufactured rods were assessed. The rods were also characterized with scanning electron microscopy (SEM) and micro-computed tomography (µ-CT). The methods are described later. 

### 2.3. Hydrolytic Degradation

The PLA70 and composite rods were cut to pieces of 7 cm in length. The samples were disinfected by rinsing with ethanol and dried before immersing them in Tris buffer (pH 7.4, Trizma, Sigma-Aldrich, Saint Louis, MO, USA) at 37 °C in a shaking incubator (100 rpm) for up to 40 weeks. Three parallel samples per time point were used. The Tris volume/rod weight ratio was maintained constant over 30:1 mL/g, and the buffer solution was changed every two weeks to keep the pH at 7.4 ± 0.2 as in [15,26]. The pH was measured at the end of the short follow-up periods (24, 48, 72 h, 1 and 2 weeks) and every two weeks before refreshing the solution at the longer follow-up periods (4, 6, 8, 10, 20, 30 and 40 weeks). The pH was measured from all three parallel samples. At each time point, the retrieved samples were rinsed with deionized water and dried in a fume cupboard for 2–3 days, followed by a week in a vacuum before the following analysis. Additionally, 5 mL of the immersion solution was collected at each time point to measure the ion concentrations.

#### 2.3.1. Water Uptake

The water uptake was calculated using the following Equation (1):(1)$$Water\;absorption=\frac{wet\;mass_{final}-dry\;mass_{final}}{dry\;mass_{final}}\times 100\%$$
where *wet mass_final_* is the wet mass recorded after the samples were blot dried and *dry mass_final_* is the dry mass recorded after drying in fume hood and vacuum. The results are presented as averages of three parallel samples.

#### 2.3.2. Ion Release

The concentrations of the ions released from the BaGs to the buffer solution were measured using inductively coupled plasma optical emission spectroscopy (ICP-OES 5110, Agilent Technologies Inc., Santa Clara, CA, USA). The samples were diluted to 1/10 dilution with deionized water and spiked with 50 µl of 70% nitric acid (Romil Ltd., Cambridge, UK) before the analysis. The following wavelengths were used in the measurements: Ca 393.366 nm, K 766.491 nm, Mg 279.553 nm, Na 589.592 nm, P 253.561, Si 250.690 nm, and Sr 216.596 nm. The results are presented as averages of three parallel samples.

#### 2.3.3. Determination of the Bioactive Glass Content

The BaG content in the composite rods was assessed by burning the matrix polymer and weighing the residual mass using a thermal analyzer (STA 449 F1 Jupiter, Netzsch, Selb, Germany). The samples (20–30 mg) were heated in alumina crucibles to 700 °C at a rate of 10 °C/min. The residual masses of the as-processed composites are given as ranges with the minimum and maximum values, and the residual masses of the composites in vitro are averages calculated from two parallel samples, given with standard deviation.

#### 2.3.4. Molecular Weight of PLA70

The degradation of the PLA70 was assessed by measuring the samples’ molecular weight with gel permeation chromatography (GPC) using a Shimadzu Prominence apparatus (Tokyo, Japan) equipped with a column set consisting of a Shodex GPC KF-G 4.6x10 mm precolumn followed by two Shodex GPC KF-806M columns (Showa Denko, Tokyo, Japan). The eluent used was chloroform flowing at a rate of 1 mL/min. The GPC was calibrated with narrow polydispersity indexpolystyrene (PS) standards in a range of 1370–2,480,000 Da. The Mark-Houwink constants used for calculations were K = 0.000545 and α = 0.73 [28] for PLA70 and K = 0.000112 and α = 0.73 [29] for the PS standards.

The GPC specimens were prepared as follows: 7.5 ± 0.15 mg of the rod was weighed and dissolved in 5 mL of chloroform overnight. All the samples were filtered through a 0.2 µm filter to remove the BaG particles before injection to the GPC. The average number-average (M_n_) and weight-average (M_w_) molecular weights of the PLA70 were calculated from the results of two parallel samples prepared. One injection per each sample was used.

#### 2.3.5. Mechanical Properties

The bending and shear properties of all the PLA70 and composite rods (2 mm × 70 mm) were measured from dry samples at room temperature using an Instron materials testing machine (Instron 4411, Instron Ltd., High Wycombe, UK). The bending properties were assessed by a three-point bending test with a 5 mm/min crosshead speed and 32 mm span. In the shear measurements, the crosshead speed was 10 mm/min. The bending and shear strength and the bending modulus were calculated using Equations (2)–(4)

Bending strength, *σ_f_* [MPa]:(2)$$\sigma_{f}=\;\frac{8\cdot \;F_{max}\cdot L}{\pi \;\cdot \;d^{3}}$$

Bending modulus, *E* [GPa]:(3)$$E=\;\frac{4\cdot \;slope\;\cdot L^{3}}{3\cdot \;\pi \cdot d^{4}}\;\cdot \;\frac{1}{1000},$$

Shear strength, *τ* [MPa]:(4)$$\tau =\;\frac{2\cdot F_{max}}{\pi \cdot d^{2}}$$
where *F_max_* = applied maximum force in testing [N], *L* = support span [mm], *d* = diameter of the rod [mm], *slope* = slope of the initial part of the stress/strain curve [N/mm]. The results are presented as averages of three parallel measurements and given with standard deviation.

#### 2.3.6. Electron Microscopy and Elemental Analysis

Scanning electron microscopy (SEM) imaging was conducted on samples as processed and after 40 weeks in vitro. Approximately 1 cm samples were cut from the rods and embedded in epoxy. The dried samples were polished by using silicon carbide grinding paper up to grit #4000. Before imaging, the samples were coated with carbon using Leica EM SCD005 cool sputter coater with CEA035 carbon evaporation accessory (Leica Microsystems GmbH, Wetzlar, Germany). The samples were imaged with FEI Quanta 200F field emission gun SEM (FEI, Hillsboro, OR, USA) with a backscatter electron detector in a high vacuum using a 20.0 kV voltage.

The elemental analysis was done using SEM/Energy-dispersive X-ray spectroscopy (SEM/EDX) Leo 1530 Gemini (Carl Zeiss AG, Oberkochen, Germany) and EDXA UltraDry (Thermo Scientific, Waltham, MA, USA). The analysis was performed inside the glass particle and at the interface of the particle and polymer. The Ca/P (Si-BaG) and (Ca + Sr)/P (P-BaG) ratios were calculated to assess changes in glass composition and possible precipitation of a CaP reactive layer. The results are presented as averages calculated from two parallel specimens with standard deviation.

#### 2.3.7. X-ray Micro-Computed Tomography

The BaG particle distribution in Si-BaG50 and P-BaG35 was assessed by X-ray micro-computed tomography (µ-CT). MicroXCT-400 (Carl Zeiss X-ray Microscopy Inc., Pleasanton, CA, USA) was used with tube voltage 80 kV and current 125 µA. The pixel size was 2.27 µm. The datasets were cropped to 1.2 mm × 1.2 mm ×1.2 mm cubes for the analysis. µ-CT visualizations were done with the Avizo 2019.3 Software (Thermo Fisher Scientific, Waltham, MA, USA).

## 3. Results and Discussion

### 3.1. Characterization of the Composites after Processing

PLA70 was successfully melt compounded with the two different bioactive glasses, Si-BaG and P-BaG. The measured BaG contents of the as-processed composites are presented in Table 3.

The target BaG contents for both glasses were 10, 30, and 50 wt.%. While the measured BaG contents were close to the nominal ones when using Si-BaG, a large deviation in glass content was observed with P-BaG composites, especially P-BaG35. The deviation is likely attributable to the higher density of the P-BaG (ρ = 2.82 g/cm^3^) compared to the Si-BaG (ρ = 2.52 g/cm^3^). The higher density, alongside a higher tendency of phosphate glasses to react to moisture [30], leads to more significant particle agglomeration and sedimentation.

During processing, the polymer degradation is mechanical (due to shear forces), thermal (due to high temperatures) or chemical (due to moisture or chemical reactions). Often the reduction in molecular weight is a mixture of these [31]. In the case of the dried, highly purified PLA70, there are no chemicals to induce chemical degradation during extrusion, so mainly thermal and mechanical degradation occur during processing. The molecular weight values of the PLA70 before and after processing and the processed composites are presented in Figure 1. Processing the PLA70 into rods decreased its molecular weight by 56% (M_n_) and 51% (M_w_), which is similar compared to our earlier study [21].

When introducing basic compounds, as BaGs, to PLA, melt compounding becomes challenging and often leads to extensive polymer degradation during processing. This is due to the addition of chemicals (i.e., OH or ions leaching from the glass) that impact the degradation behavior. Blaker et al. reported, and we confirmed in our processing trials, that extruding composites of S53P4 glass and PDLLA induced bubble formation, amber color, sweet odor, and over 85% decrease in the polymer molecular weight at a reasonable processing temperature of 150 °C, suggesting extensive degradation of the polymer. It was concluded that the degradation was due to both high shear forces and the presence of the BaG. [32] The high surface energy in some Si-BaGs [33] leads to water absorption on their surface to form Si-OH, which can accelerate PLA degradation in extrusion. Partial glass dissolution in the polymer melt during processing may also induce more chemical stress to the polymer.

A less reactive yet bioactive Si-BaG was used for the melt compounding in the current study. It was hypothesized that a less reactive glass would induce less degradation during processing. The molecular weight of PLA70 in the Si-BaG composites decreased by 71–77% (M_n_) and 72–79% (M_w_) during processing, which is less compared to the drop in M_w_ with S53P4, but more compared to our previous study [21]. This high drop in molecular weight was not anticipated. The glass in this study had a higher Na_2_O content compared to the one in [21], which may account for a slightly higher reactivity. In addition, the extruder configuration was different compared to the processing set-up in [21], along with a smaller die. In the case of the Si-BaG composites, the degradation is most probably a mixture of the effects of mechanical, chemical and thermal stresses. It appears that with the Si-BaGs, the more reactive the glass is, the more readily the excessive degradation of the polymer occurs during processing, as also noted in [34]. In addition, the shear forces in extrusion increase the chemical stress due to surface reactions between PLA and BaG particles [32].

The use of P-BaG, on the other hand, was not associated with a significant change in the PLA70 viscosity, indicating that the glass composition drastically affects the outcome of the PLA70 stability. Adding P-BaG into the composites led to a similar decrease in M_n_ (52–60%) and M_w_ (47–54%) than when processing the PLA70. This suggests that the degradation of these composites was mainly thermal. Despite the P-BaG being more reactive than the Si-BaG, no significant degradation during processing was noticed with the composites with P-BaG. Therefore, the dissolution mechanism of the BaGs affects the properties of the PLA70 melt.

The initial mechanical properties of the PLA70 and composite rods are presented in Figure 2.

The initial bending strength of PLA70 was 116 ± 13 MPa, shear strength 46 ± 0.2 MPa, and bending modulus 3.5 ± 0.6 GPa, similar to our previous study [35]. Adding BaGs decreased the bending strength by 3–43% (Si-BaG) and 12–45% (P-BaG), and shear strength by 4–35% (Si-BaG) and 2–37% (P-BaG) compared to the PLA70. The addition of Si-BaG resulted in similar (0–9% lower) modulus values compared to PLA70, and adding P-BaG decreased the modulus only slightly (9–23%), which indicates that the material became more ductile. Such behavior has also been reported earlier [21]. The results are of interest since the change in the mechanical properties did not correlate to the glass composition or polymer degradation during processing but rather with the glass content.

It should be noted that higher glass content resulted in lower mechanical strength, but the change was less visible in the modulus results. Decreasing mechanical properties in twin-screw extruded composites of PLA and BaG were also reported in [26,36]. In these studies, 30–50% of Si-BaG in PLA decreased the initial bending strength by 28–38% and shear strength by 18–20%. The decrease was attributed to the lack of chemical bonding between the glass and the polymer and the formation of small pores during the extrusion.

The composition of the BaG introduced into the PLA70 distinctly impacts the PLA70 molecular weight. Si-BaG particles tend to accelerate the PLA70’s thermal degradation, whereas the P-BaG particles do not seem to interact significantly with the PLA70 during processing. This difference is of interest as it shows that tailoring the Si-BaG composition can help maintain a high polymer molecular weight and, thus, the polymer integrity. It is also important to point out that the composites’ mechanical properties are more closely linked to the glass content than the glass composition.

### 3.2. Degradation of the PLA70 and the Composites

#### 3.2.1. pH of the Buffer Solution 

The pH of the buffer solution with PLA70 and the composites are presented in Figure 3.

The PLA70 did not induce any notable changes in the pH during the studied dissolution period of 40 weeks, which agrees with our previous studies [26]. The Si-BaG composites exhibit an increase in pH over the first two weeks of immersion (inset in Figure 3a). This pH increase is more pronounced for the composites with higher glass content than those with less glass. Such pH change is related to the dissolution of the glass, which involves the exchange of Na^+^ and K^+^ ions with H^+^ and release of alkaline and alkaline earth ions into the solution [7,37], also seen in our previous work [26]. The higher the glass content, the more prominent is the pH change and the release of ions. On the other hand, the solutions containing the P-BaG composites exhibited a decrease in pH (inset in Figure 3b), especially at the highest glass content. The decrease in pH is due to the release of phosphate ions forming phosphoric acid [38,39].

#### 3.2.2. Water Absorption of the Specimens

The water absorption of the studied materials is shown in Figure 4.

The water absorption of the PLA70 alone was negligible (2 wt.%) over the 40 weeks. PLA70 is a hydrophobic material, and these results are in line with previous research [15,22,26]. The Si-BaG10, Si-BaG30, and Si-BaG50 composites absorbed up to 32, 169, and 222 wt.% of water in 40 weeks, respectively. The absorption increased with increasing glass content. The Si-BaGs are hydrophilic and absorb water to form a silica gel layer when they dissolve. In our previous studies, a self-reinforced composite of PLA70 and similar Si-BaG absorbed water up to approximately 30% (composites of ~20 wt.% glass) and 60% (50 wt.% glass content) in 40 weeks [15,26]. 

The water absorption of the P-BaG composites was much lower at 40 weeks in vitro (53, 80, and 93 wt.% with P-BaG10, P-BaG25, and P-BaG35, respectively) compared to the composites with Si-BaG. However, it is important to note that the initial water absorption (up to 10 weeks) of the P-BaG composites was significantly faster than for the Si-BaG composites. This difference can be related to the dissolution rate of the various glasses. Indeed, P-BaGs in the metaphosphate structure are typically dissolving faster than Si-BaGs [40]. Also, it is noteworthy that the dissolution mechanism is different. Si-BaGs dissolve in a non-congruent manner, whereas the P-BaGs dissolve congruently [7,40]. The non-congruent dissolution of the Si-BaG may explain the slow initial increase in water absorption. H^+^ / Na^+^ exchange and the release of alkaline earth and soluble silica will lead to slow water retention. When the silica gel forms and grows thicker with time, more water will be retained in the composite structures. Also, the faster dissolution of the P-BaG is partially confirmed by the plateau after 20 weeks. This plateau may indicate that the glass has dissolved.

#### 3.2.3. Ion Release from the Glasses

The glass dissolution was tracked using ICP-OES to quantify the concentration of ions in the solution. The ion concentrations in the buffer solution as a function of time are presented in Figure 5 (Si-BaG) and Figure 6 (P-BaG).

The buffer solution was refreshed at least every two weeks to avoid saturation, and thus the results are shown as a cumulative release. The concentration of each element increased with time and with increasing glass content in all the composites, as expected, indicating that the ions leach out from the glass and diffuse through the polymer matrix. The other ions’ release except for potassium in Si-BaG and phosphorus in P-BaG was reasonably constant. A linear release is regarded as positive, as it is predictable, controllable, and enables incorporating new therapeutic ions into the glass. 

In theory, as previously mentioned, the Si-BaG dissolution starts with a release of alkaline and alkaline earth ions [7,37]. The release of soluble silica then follows it. Therefore, as expected, all ions are released at a constant rate except for the silicon. However, the release of potassium was not expected to slow down and may indicate that these ions remained trapped within the polymer matrix.

In the dissolution of P-BaG, the amount of phosphorus in the solution with the two highest contents of P-BaG is increasing rapidly until 15 weeks and then slowing down. This rapid initial ion release is a sign of the higher reactivity of the P-BaG and agrees with the water absorption results. It should be noted that the dissolution of this glass was studied earlier and established to be congruent [38,40]. It is unlikely that the glass dissolution mechanism changes when embedded into the PLA70 matrix. Therefore, it is conceivable that the shape of the PO_4_^3−^ release indicates ion diffusion through the polymeric matrix. The large ionic radius and the charge of the phosphorus ion delays its release in the surrounding solution.

Surprisingly, the sodium and calcium releases are similar between different glass contents, whereas strontium and phosphorus increase with increasing glass content. Furthermore, the ion release rate seems to correlate with the cations’ ionic radius, with the Sr^2+^ (radius 132 pm) being released at a slower rate than Ca^2+^ and Na^+^ (114 and 116 pm, respectively). Finally, while most of the Na is being released in solution, Ca and Sr appeared to be released in smaller amounts than the glass stoichiometry. This finding, alongside the saturation of the phosphorus release, are consistent with the precipitation of a Sr substituted CaP layer as evidenced in this glass composition [27,41,42]

#### 3.2.4. Bioactive Glass Content in the Composites

The glass particle contents in the composites as a function of immersion time are presented in Figure 7.

The glass amount in Si-BaG10 seems to have stayed constant during the immersion time. In Si-BaG30 and Si-BaG50, 25–42 wt.% glass was left after 40 weeks *in vitro*, which corresponds to 12–20% drops compared to the initial glass content. The decrease in the glass content confirms the dissolution of the glass.

The P-BaG seemed to dissolve much faster compared to the Si-BaG. The residual mass in the P-BaG composites decreased to 6–15 wt.% over 40 weeks, corresponding to 4–60% drops compared to the initial glass content. The decrease in the lowest content was much smaller compared to the drops in the two higher contents, which were very similar (55 and 60%). However, it should be kept in mind that if a precipitate forms, the residue corresponds to the remaining glass and the reactive layer.

#### 3.2.5. Molecular Weight Retention of PLA70

The degradation of the polymer matrix in the composites was assessed by measuring its molecular weight using GPC. The molecular weight retention of the materials during the in vitro hydrolysis is presented in Figure 8. The molecular weights of PLA70 in all the sample types started to decrease right from the start of the hydrolysis. The molecular weight of PLA70 decreased by 55% (M_n_) and 64% (M_w_) in 40 weeks in vitro, in agreement with our previous research [15], where self-reinforced and sterilized PLA70 retained less than 20% of its initial M_w_ in 40 weeks of hydrolysis. Interestingly, in the article by Houaoui et al., the M_w_ of PLA70 did not notably decrease during hydrolysis of 10 weeks. However, their rods’ degradation might have been slower because of a slightly larger rod diameter [21].

Adding either BaG in PLA70 accelerated its degradation. The Si-BaG composites had 9–11% (M_n_) and 9–12% (M_w_), and the P-BaG composites 33–40% (M_n_) and 26–36% (M_w_) left of their initial molecular weight after 40 weeks. The P-BaG seems to have a lesser impact on the degradation rate of PLA70 in vitro compared to the Si-BaG composites, despite the higher reactivity of the P-BaG. In the Si-BaG composites, the decrease in M_n_ and M_w_ are almost identical, but with the P-BaG, the decrease was more prominent in M_w_. The more prominent degradation of PLA70 in vitro in the Si-BaG composites is probably related to the greater extent of the polymer’s thermal degradation during processing. Alternatively, the increase in pH at the glass/polymer interface associated with the dissolution of the Si-BaGs may have led to alkaline hydrolysis of ester bonds in extrusion [43]. All the molecular weight distribution curves were monomodal in agreement with [15,24], suggesting a uniform polymer chain length distribution.

#### 3.2.6. Mechanical Properties

The bending and shear strength retention and bending modulus of the materials in vitro are presented in Figure 9.

The PLA70 did not lose its mechanical properties during the 40 weeks in hydrolysis, despite the decrease in molecular weight. In earlier studies [15,26], self-reinforced PLA70 lost approximately 20% of its shear strength in 20 weeks, maybe because of losing the self-reinforcement due to molecular relaxations during the hydrolysis and sterilization. Unlike PLA, the composites lost their mechanical properties quite linearly already from the start of the hydrolysis. The Si-BaG composites retained 3–23% of their initial bending strength after 40 weeks in vitro. Si-BaG10 had 48% of its initial shear strength left after 40 weeks, and Si-BaG30 and Si-BaG50 had 44 and 27% left after 30 weeks (on 40 weeks, the extent of the degradation of Si-BaG30 and Si-BaG50 prohibited the shear testing). The Si-BaG composites had 6–47% of their bending modulus left at 40 weeks. The composites lost their mechanical integrity slightly faster than in our previous study [15], where the composites with 20% and 30% Si-BaG had 60% of their initial shear strength left after 40 and 30 weeks in vitro, respectively. The shear strength retention of Si-BaG50 corresponds with our previous results [15]. It could be noted that the more Si-BaG there was in the structure, the faster was the decrease in mechanical properties, most probably due to the accelerated degradation caused by processing or due to the excessive water absorption of these composites.

The P-BaG composites had 43–47% of their initial bending strength, 62–72% of their initial shear strength, and 19–42% of their bending modulus left after 40 weeks. The P-BaG25 and P-BaG35 lost their bending strength slightly faster compared to P-BaG10. However, in the shear test, there was no correlation between glass content and loss of shear strength. The loss of mechanical properties is quite similar to the results of Felfel et al., where the composites of PLA and P-BaG lost around 45–50% of their initial flexural properties in 8 weeks in vitro [24].

#### 3.2.7. Scanning Electron Microscopy and Elemental Analysis

The SEM images of Si-BaG50 and P-BaG35 are presented in Figure 10.

Figure 10a shows that the glass particles in the Si-BaG composite were evenly distributed after processing. After 40 weeks in vitro, the sample’s diameter had notably increased, and the structure was much more porous, but there was still some glass left (Figure 10e). The swelling was also seen in our previous study, where most self-reinforced composites lost their shape between approximately 18 and 30 weeks in vitro [15]. Figure 10f shows the darker area at the glass particles’ surface, which indicates the glass’s reaction with the aqueous solution.

In P-BaG35 before hydrolysis (Figure 10c), the particles seem to be smaller compared to Si-BaG50, even though both glasses were sieved to the same particle size range. The particles might have broken down due to the mechanical forces during processing, or small particles have been agglomerated with the larger particles when crushing the glass. Figure 10g shows the cross-section of the P-BaG composite after 40 weeks in vitro. The P-BaG composite diameter also grew, although less compared to the Si-BaG composites, as expected from the lower water absorption. Nearly all the P-BaG has dissolved, and only hollow particles are left in the structure (Figure 10h).

Elemental analysis with EDX was carried out to analyze how the glasses composition changed in vitro. The ratios of Ca/P (Si-BaG) and (Ca + Sr)/P (P-BaG) before and after ten weeks in vitro are presented in Table 4. An example of EDX analysis is presented in Appendix A.

First, it is noteworthy that the ratios before immersion align with the glasses nominal composition (Table 1). The Ca/P ratio in the Si-BaG after ten weeks in vitro is similar between the particles’ core and the edge, indicating that the glass has not yet undergone significant reaction with the medium. However, the ratio is lower than in the nominal glass composition, indicating that the first stage of the glass dissolution had initiated. At 40 weeks, the darker areas seen on the glass appeared to be depleted in alkaline and alkaline earth, indicating the start of the formation of a silica reach layer (Figure 10f). The buffer solution was probably changed so often, and the glass dissolution was so slow that no saturation in Ca and P occurred, thus preventing the precipitation of a reactive layer. In P-BaG, the (Ca+Sr)/P ratio at the center of the particles remained similar to the nominal glass composition, consistent with the congruent dissolution of phosphate glasses. On the edges, a distinctive reactive layer appeared with (Ca+Sr)/P ~1. The layer’s composition corresponds to a dicalcium diphosphate dihydrate layer, previously reported to form at the surface of this glass [44].

#### 3.2.8. Structural Analysis

The cross-sectional µ-CT images of Si-BaG50 and P-BaG35 at 0, 20, and 40 weeks in vitro are gathered in Figure 11.

From the figure, we can see that the glass particles are irregularly shaped but relatively evenly distributed in both composites (Figure 11a,d). There is also some porosity in the PLA70 matrix already after processing, explaining the low mechanical properties of the composites. There are not notably less Si-BaG particles after 20 weeks in vitro, but some darker areas are visible at the edges of the particles as signs of degradation. After 40 weeks (Figure 11c), the number of particles has notably decreased, and the remaining particles show more signs of degradation than at 20 weeks. In the P-BaG composite (Figure 11d–f), nearly all the glass had dissolved before 20 weeks in vitro, and no difference can be seen between 20 and 40 weeks in vitro. Only small, hollow particles can be seen, and the structure appears very porous. In Figure 12, 3D images of the composites are presented. The images also show that P-BaG dissolved at a higher rate than Si-BaG, and there was almost no glass left already at 20 weeks in vitro.

## 4. Conclusions

PLA70 was successfully melt-compounded with two different bioactive glasses, Si-BaG and P-BaG. Si-BaG was found to induce more chemical degradation of PLA70 during processing, whereas P-BaG did not. The PLA70 and composites were studied for 40 weeks in vitro. The addition of BaGs to PLA70 accelerated the polymer’s in vitro degradation, resulted in poorer mechanical properties, increased porosity and hydrophilicity of the materials. The glass composition was found to influence the thermal and hydrolytic degradation rate of the polymer. However, the loss of the mechanical properties was more linked to the glass content than the composition. Surprisingly, the more reactive P-BaG was found to have a lesser impact on polymer degradation than the more stable Si-BaG, both in processing and in vitro. When melt compounded with Si-BaG, the increased polymer degradation is believed to be due an increased chemical stress by an increase in pH at the glass/polymer interface, leading to the alkaline hydrolysis of the ester bonds.

Both glasses were found to dissolve through the polymer matrix. Most of the P-BaG had already dissolved before 20 weeks, while there was still Si-BaG after 40 weeks. All the composites exhibited a linear, predictable ions release, which enables incorporating therapeutic ions into the glasses. Tuning the composition of the Si-BaG opens possibilities to make composites with slower polymer degradation. The P-BaG composites showed great promise despite the reactivity of the glass.

This research introduces new composites of polymer and bioactive glasses with tailorable ion release for applications in bone regeneration and fixation. The devices would release therapeutic ions to induce osteogenesis, vascularization, or antimicrobial properties without additional chemical substances.

## Figures and Tables

**Figure 1 materials-14-00667-f001:**
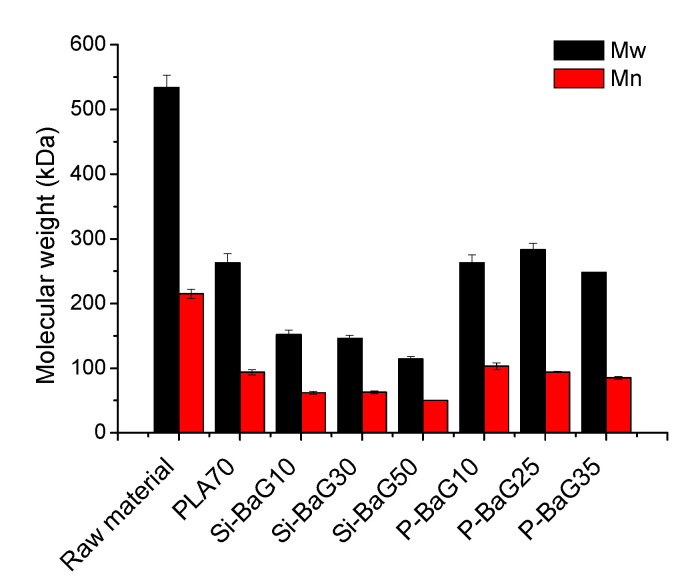
The number-average (M_n_) and weight-average (M_w_) molecular weights of the raw material and the as-processed PLA70 and composites.

**Figure 2 materials-14-00667-f002:**
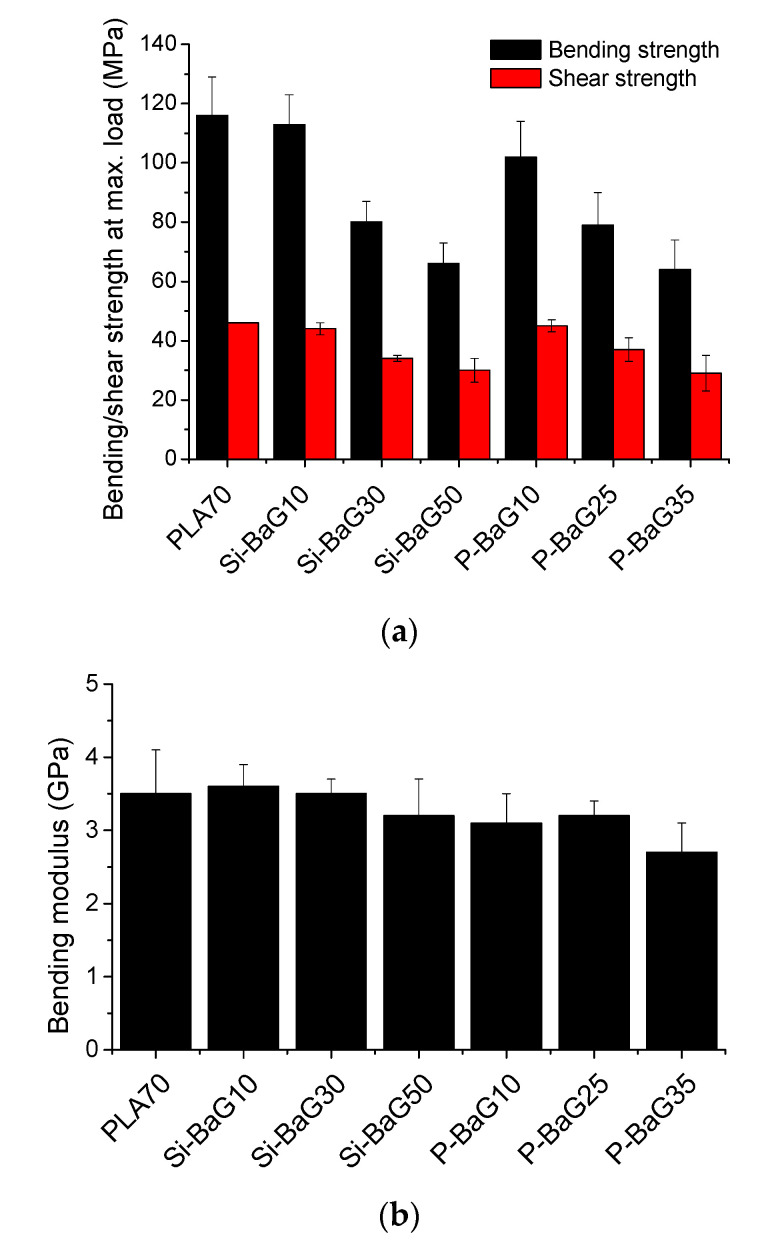
The initial (**a**) bending and shear strength and (**b**) bending modulus of PLA70 and its composites with Si-BaG and P-BaG.

**Figure 3 materials-14-00667-f003:**
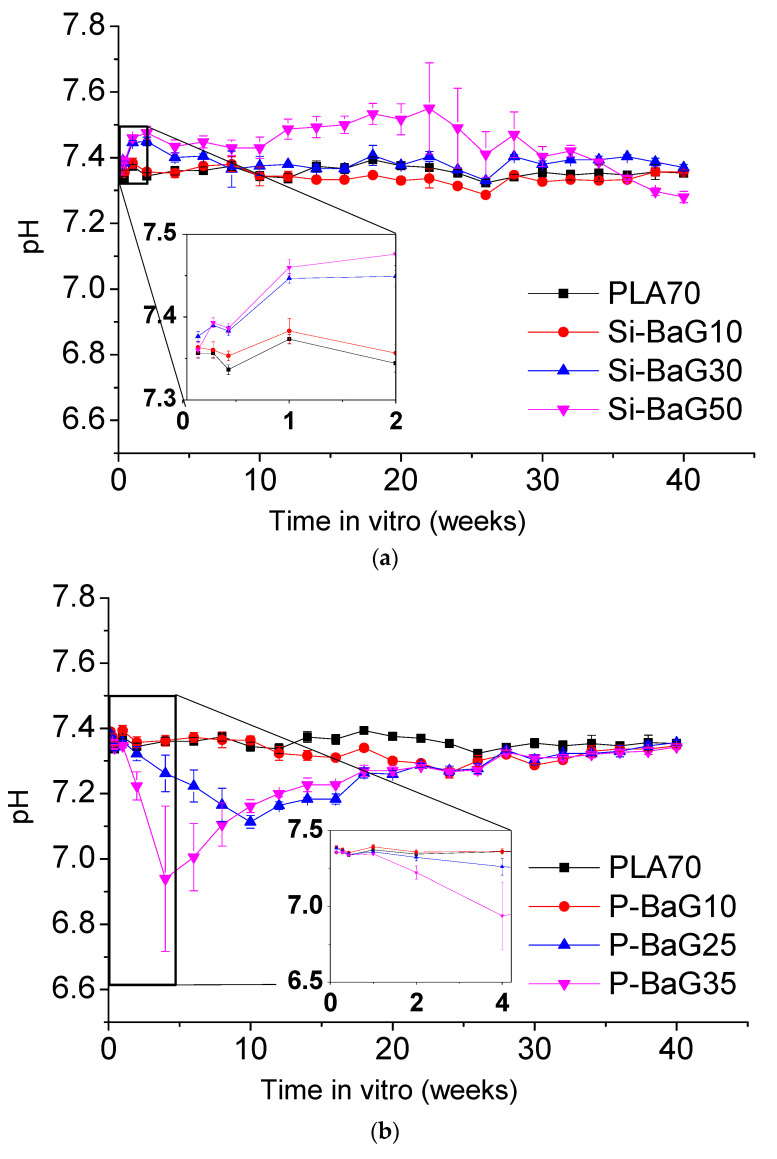
pH of the buffer solution with PLA70 and its composites with (**a**) Si-BaG and (**b**) P-BaG during 40 weeks in Tris.

**Figure 4 materials-14-00667-f004:**
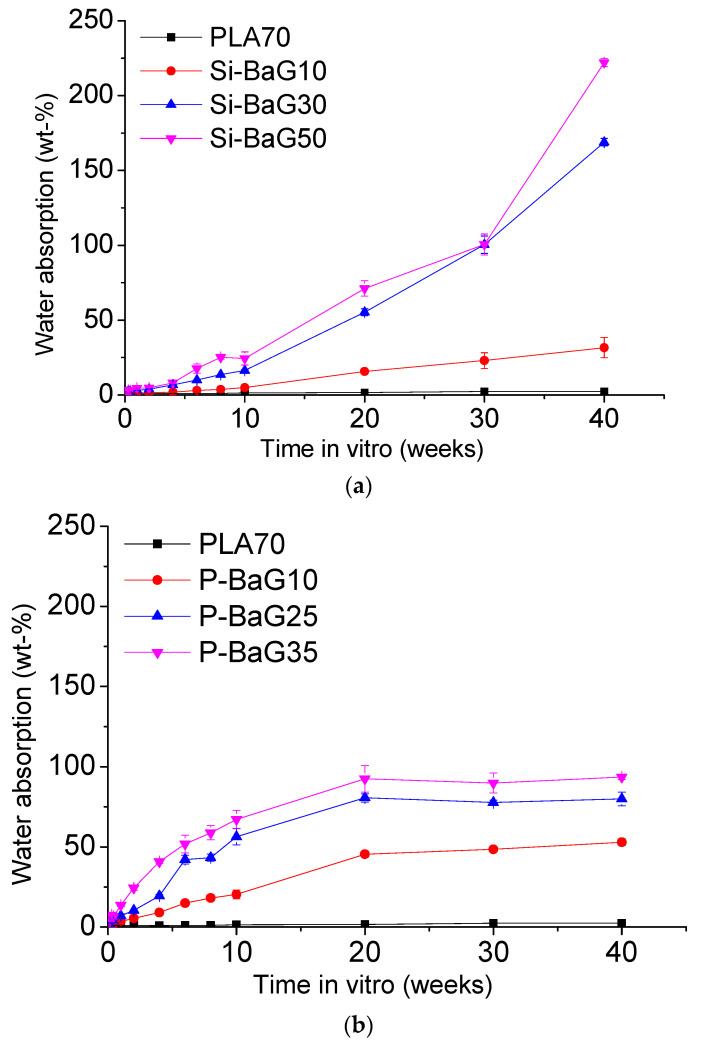
Water absorption of (**a**) Si-BaG and (**b**) P-BaG composites in Tris over 40 weeks.

**Figure 5 materials-14-00667-f005:**
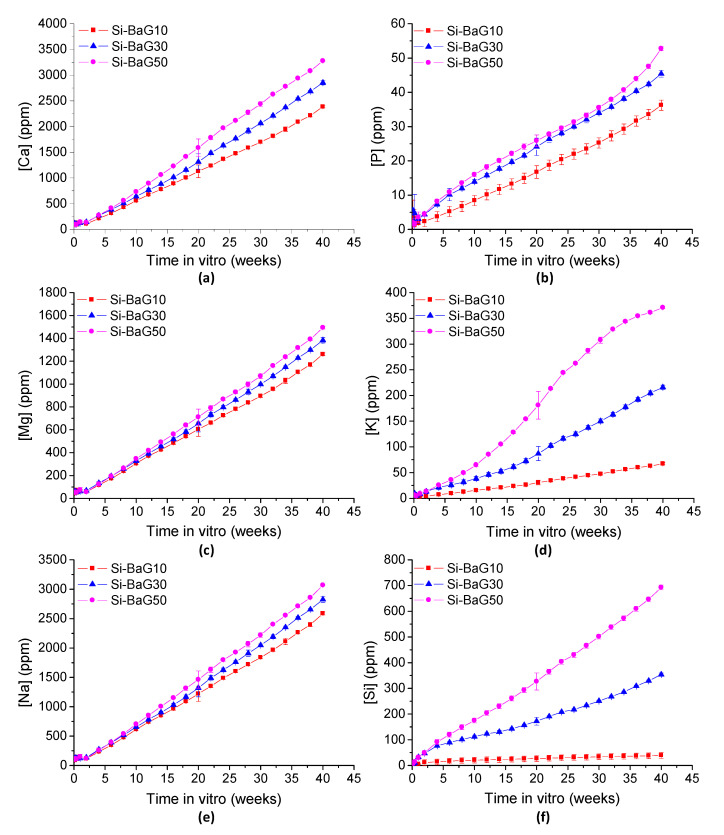
The (**a**) calcium, (**b**) phosphorus, (**c**) magnesium, (**d**) potassium, (**e**) sodium, and (**f**) silicon ions released from the Si-BaG composites into the Tris solution over 40 weeks.

**Figure 6 materials-14-00667-f006:**
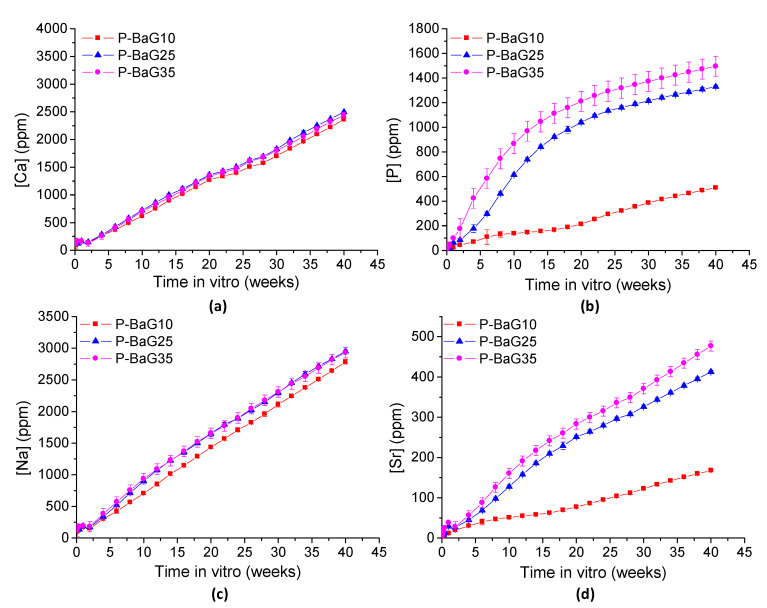
The (**a**) calcium, (**b**) phosphorus, (**c**) sodium, and (**d**) strontium ions released from the P-BaG composites into the Tris solution over 40 weeks.

**Figure 7 materials-14-00667-f007:**
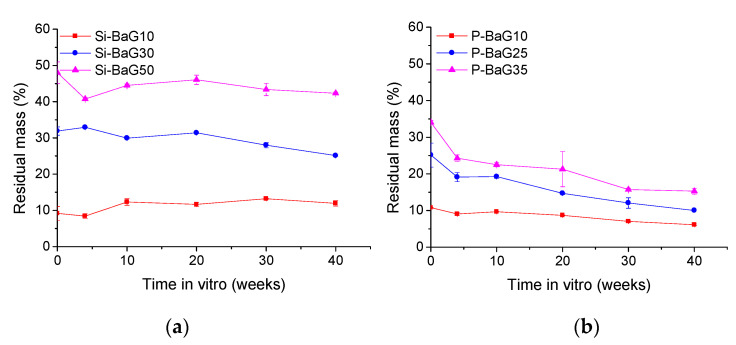
Changes in the BaG content in (**a**) Si-BaG and (**b**) P-BaG composites during the immersion of 40 weeks in Tris.

**Figure 8 materials-14-00667-f008:**
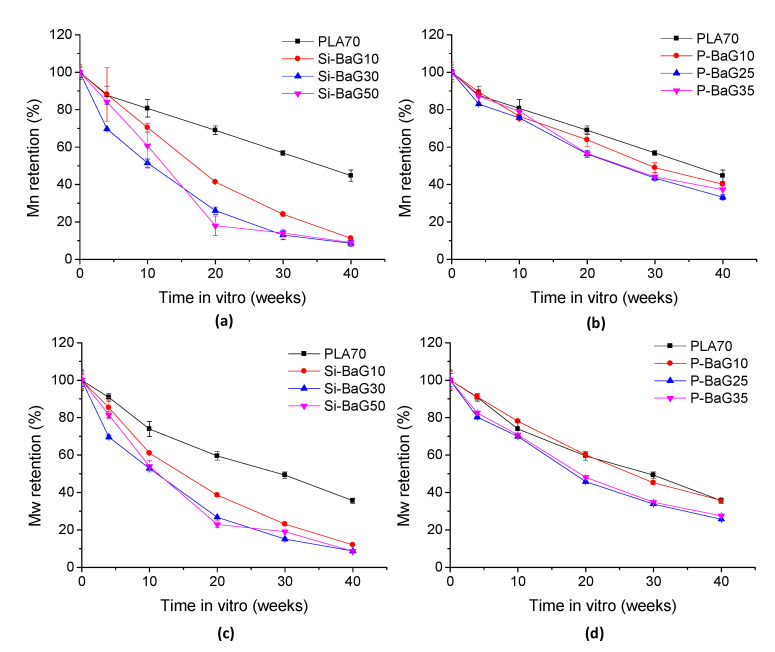
M_n_ retention of PLA70 with (**a**) Si-BaG and (**b**) P-BaG and M_w_ retention of PLA70 with (**c**) Si-BaG and (**d**) P-BaG during the in vitro hydrolysis of 40 weeks in Tris.

**Figure 9 materials-14-00667-f009:**
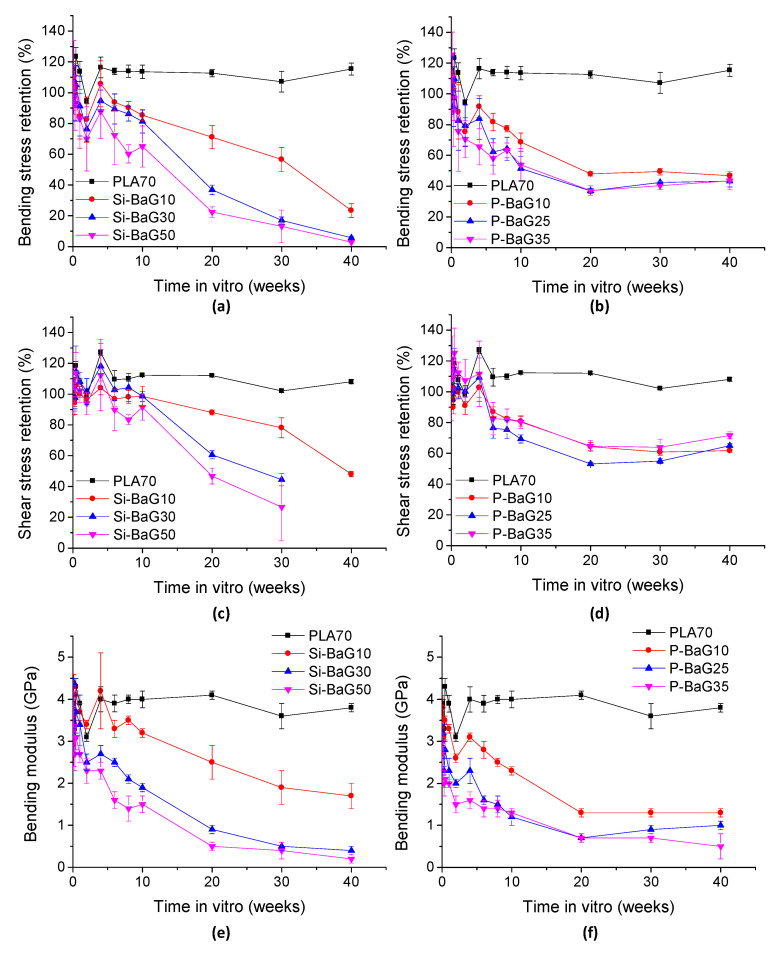
Bending and shear strength retention and bending modulus of PLA70 and the composites with (**a**), (**c**), (**e**) Si-BaG and (**b**), (**d**), (**f**) P-BaG over a period of 40 weeks in vitro.

**Figure 10 materials-14-00667-f010:**
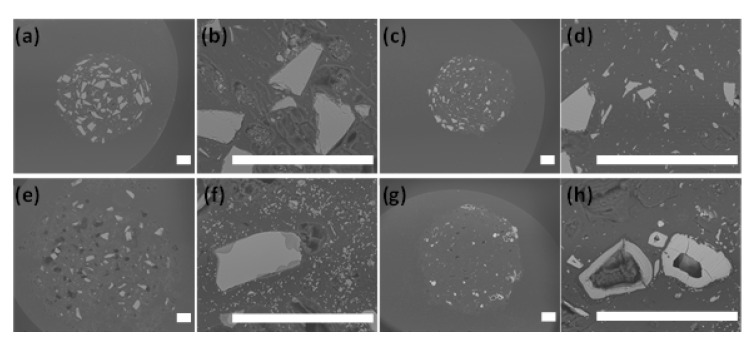
Scanning electron microscopy cross-sectional images of Si-BaG50 (**a**,**b**) before in vitro hydrolysis, (**e**,**f**) after 40 weeks in vitro and P-BaG35 (**c**,**d**) before hydrolysis and (**g**,**h**) after 40 weeks *in vitro*. Scale bar 200 µm.

**Figure 11 materials-14-00667-f011:**
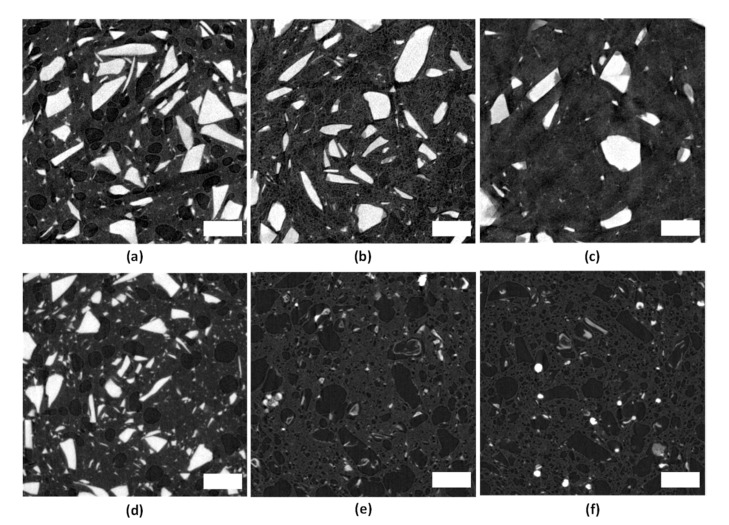
µ-CT images of PLA-BaG composites. Top row from left to right: Si-BaG50 (**a**) 0, (**b**) 20 and (**c**) 40 wk and bottom row from left to right: P-BaG35 (**d**) 0, (**e**) 20 and (**f**) 40 wk. Scale bar 200 µm.

**Figure 12 materials-14-00667-f012:**
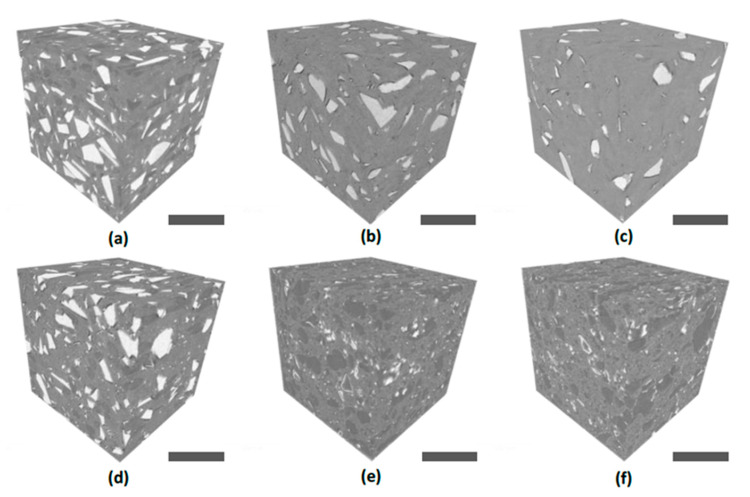
3D images of PLA-BaG composites. Top row: Si-BaG50 (**a**) 0, (**b**) 20 and (**c**) 40 wk and bottom row: P-BaG35 (**d**) 0, (**e**) 20 and (**f**) 40 wk. Scale bar 500 µm.

**Table 1 materials-14-00667-t001:** Compositions of the bioactive silicate-based (Si-BaG) and phosphate-based (P-BaG) bioactive glasses in mol%.

Glass	SiO_2_	P_2_O_5_	CaO	Na_2_O	MgO	K_2_O	SrO
Si-BaG	54.6	1.7	22.1	8.0	7.7	5.9	-
P-BaG	-	50.0	20.0	10.0	-	-	20.0

**Table 2 materials-14-00667-t002:** The processing conditions of the PLA70 and the composites.

Parameters	PLA70	Si-BaG10	Si-BaG30	Si-BaG50	P-BaG10	P-BaG25	P-BaG35
Temperature (°C)	178–203	172–198	169–195	169–196	180–206	193–217	192–214
Pressure (psi)	750–850	470–500	630–730	600–630	1300	790–800	860–900

**Table 3 materials-14-00667-t003:** The glass contents of the as-processed composites.

Material	Si-BaG10	Si-BaG30	Si-BaG50	P-BaG10	P-BaG25	P-BaG35
Glass content (wt.%)	7.3–11.1	30.7–33.1	45.0–51.0	10.7–10.9	21.8–28.4	33.6–34.6

**Table 4 materials-14-00667-t004:** The calculated ratios of the Si-BaG (Ca/P) and P-BaG ((Ca + Sr)/P) composites in atom % after ten weeks in vitro compared to the glass only before in vitro.

Sample	Ca/P or (Ca+Sr)/P Ratios before In Vitro Hydrolysis	Ca/P or (Ca+Sr)/P Ratios inside the Glass Particle, Ten Weeks In Vitro	Ca/P or (Ca+Sr)/P Ratios on the Interface of BaG and PLA70, Ten Weeks In Vitro
Si-BaG10	6.70 ± 0.50	5.76 ± 0.03	5.79 ± 0.04
Si-BaG30	6.70 ± 0.50	4.41 ± 0.04	4.30 ± 0.41
Si-BaG50	6.70 ± 0.50	4.11 ± 0.07	4.21 ± 0.36
P-BaG10	0.35 ± 0.01	0.37 ± 0.03	1.06 ± 0.09
P-BaG25	0.35 ± 0.01	0.36 ± 0.05	0.81 ± 0.07
P-BaG35	0.35 ± 0.01	0.34 ± 0.02	0.90 ± 0.20

## Data Availability

Data available on request.

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
