# Peer review of "Impact of Glass Composition on Hydrolytic Degradation of Polylactide/Bioactive Glass Composites"

_materials, 2021, doi:10.3390/ma14030667_

Round 1

Reviewer 1 Report

This is an interesting and well-presented study. However there are some issues that need to be addressed before being considered for publication.

  1. page 2, 3rd paragraph: PLDLA needs to be referred with its whole name and the abbreviation in parenthesis
  2. In the materials section for the hydrolysis test the authors write the following: “The TRIS volume/rod weight ratio was maintained constant over 30:1 ml/g. The buffer solution was changed every two weeks”. Why did they choose this volume to rod weight ratio? Why did they choose to change the medium every two weeks? Are there any references to support their choice?
  3. Please add the standard deviation of the glass contents in Table 3
  4. In page 6, at the end of the second paragraph, the authors state: “…high intrinsic water content in some bioactive glasses”. What do the authors mean by that? What is the intrinsic water?
  5. In page 6, at the end of the second paragraph the authors state that: “It appears that with the Si-BaGs, the more reactive the glass is, the more readily the excessive thermal degradation of the polymer occurs”. However, in a few sentences below they say: ‘Despite the P-BaG being more reactive than the Si-BaG, no significant thermal degradation was noticed with the composites with P-BaG’. It seems to me that other reason rather than glass reactivity is responsible for the different behavior. Could the authors comment on this?
  6. Page 7, 2nd paragraph: the authors state that: “Decreasing mechanical properties in twin-screw extruded composites of PLA and BaG were also reported in [26,32]. They reported that 30–50% of SiBaG in PLA decreased the initial bending strength by 28–38% and shear strength by 18–20%” . The beginning of the second sentence (“they reported”) is referred to references 16,32, however  they should mention the respective authors or rephrase perhaps by writting “in these studies, 30-50% of SiBaG in PLA decreased the initial bending strength by 28–38% and shear strength by 18–20%”
  7. I think it is important to perform appropriate statistical analysis to see what difference is important regarding the mechanical properties.
  8. The different rate of ions dissolution of phosphorous and strontium in P-BaGs is justified by the authors by the following: “This phenomenon could be assigned to the saturation of phosphorus, which tends to indicate that precipitation of a reactive layer occurs”. If this is the case, why the amount of calcium was so stable? It should be also saturated if a calcium phosphate reactive layer had formed. Could the authors give an explanation for this? Can they provide an FTIR spectrum or XRD pattern to prove their statement?
  9. In table 4 the authors present the calculated ratios of the Si-BaG (Ca/P) and P-BaG ((Ca+Sr)/P) composites in atom-% after ten weeks in vitro compared to the glass only before in vitro. From how many samples was this analysis performed? Was it spot or area analyses? Can they provide micrographs showing the analyzed areas?
  10. In general I believe that more detailed discussion on the findings is needed to reach the standard of the journal. Also English language needs improvements.

Author Response

Answers to reviewers

First of all, I (on behalf of all co-authors), would like to thank the reviewers for their constructive comments.

The authors have attempted to answer every query by the reviewers as marked in red in the text.

Reviewer 1

This is an interesting and well-presented study. However there are some issues that need to be addressed before being considered for publication.

  1. page 2, 3rd paragraph: PLDLA needs to be referred with its whole name and the abbreviation in parenthesis

The whole name of the polymer and the parentheses were added.

  1. In the materials section for the hydrolysis test the authors write the following: “The TRIS volume/rod weight ratio was maintained constant over 30:1 ml/g. The buffer solution was changed every two weeks”. Why did they choose this volume to rod weight ratio? Why did they choose to change the medium every two weeks? Are there any references to support their choice?

The text was modified as follows: The TRIS volume/rod weight ratio was maintained constant over 30:1 ml/g and the buffer solution was changed every two weeks to keep the pH at 7.4 ± 0.2 as in [15,26].

  1. Please add the standard deviation of the glass contents in Table 3

The authors chose to present the glass contents as ranges because the standard deviation could not be calculated from two parallel samples.

  1. In page 6, at the end of the second paragraph, the authors state: “…high intrinsic water content in some bioactive glasses”. What do the authors mean by that? What is the intrinsic water?

The part was rewritten and a reference was added for clarification: The possible reasons for the drop in the Mw of PLA70 could be partial glass dissolution in the polymer melt or the high surface energy in some Si-BaGs [32], leading to water absorption on their surface to form Si-OH.

  1. In page 6, at the end of the second paragraph the authors state that: “It appears that with the Si-BaGs, the more reactive the glass is, the more readily the excessive thermal degradation of the polymer occurs”. However, in a few sentences below they say: ‘Despite the P-BaG being more reactive than the Si-BaG, no significant thermal degradation was noticed with the composites with P-BaG’. It seems to me that other reason rather than glass reactivity is responsible for the different behavior. Could the authors comment on this?

The different reasons for the thermal degradation were discussed in the text and they included the high surface energy of the Si-BaG and partial glass dissolution in the polymer melt. A reference was added to support the claim.

  1. Page 7, 2nd paragraph: the authors state that: “Decreasing mechanical properties in twin-screw extruded composites of PLA and BaG were also reported in [26,32]. They reported that 30–50% of SiBaG in PLA decreased the initial bending strength by 28–38% and shear strength by 18–20%” . The beginning of the second sentence (“they reported”) is referred to references 16,32, however  they should mention the respective authors or rephrase perhaps by writting “in these studies, 30-50% of SiBaG in PLA decreased the initial bending strength by 28–38% and shear strength by 18–20%”

Thank you. The text was rephrased according to the suggestions.

  1. I think it is important to perform appropriate statistical analysis to see what difference is important regarding the mechanical properties.

The authors agree. However, the sample size here (3 parallel samples per time point) did not allow a reliable statistical analysis to be made.

  1. The different rate of ions dissolution of phosphorous and strontium in P-BaGs is justified by the authors by the following: “This phenomenon could be assigned to the saturation of phosphorus, which tends to indicate that precipitation of a reactive layer occurs”. If this is the case, why the amount of calcium was so stable? It should be also saturated if a calcium phosphate reactive layer had formed. Could the authors give an explanation for this? Can they provide an FTIR spectrum or XRD pattern to prove their statement?

The author would like to thank the reviewer for pointing this out. It appears that the PO43- follows a diffusion process and that might be attributable to its larger ionic size compared the one of the cations. Furthermore, a congruent dissolution from the phosphate glass is expected. However, the amounts of Ca and Sr are much lower than Na, when taking the stoichiometry of the glass into consideration.
Therefore, the profile of the phosphorus release, alongside with the lower release of the Ca and Sr seem to be in agreement with the precipitation of a reactive layer (Sr-substituted CaP layer)

The text has been modified to reflect this argumentation.

  1. In table 4 the authors present the calculated ratios of the Si-BaG (Ca/P) and P-BaG ((Ca+Sr)/P) composites in atom-% after ten weeks in vitro compared to the glass only before in vitro. From how many samples was this analysis performed? Was it spot or area analyses? Can they provide micrographs showing the analyzed areas?

The elemental analysis was performed on two parallel samples, as stated in the materials and methods (p. 5). The analysis was performed on a spot.

A supplementary figure has been added to show the formation of the reactive layer in P-BaG composites

  1. In general I believe that more detailed discussion on the findings is needed to reach the standard of the journal. Also English language needs improvements.

The authors hope that the answers to the reviewers clarified and improved the discussion/interpretation of the results. However, if not, please specifically indicate which part should be improved and that will be done in the next review if necessary.

Reviewer 2 Report

In this manuscript the authors investigated the degradation behavior of PLA-BaG composites. They chose a silicate and phosphate BaG to determine the influence of the BaG composition on the overall degradation behavior of the composites.

The topic is of great interest, the investigations are conducted thoroughly and extensively, ranging from mechanical properties, to degradation behavior of the polymer as well as ion release from the glasses and visual characterization during and after degradation by SEM/EDX and µCT.

The manuscript is well written, the results are properly presented and discussed. This manuscript can be published as is.

Author Response

Reviewer 2

In this manuscript the authors investigated the degradation behavior of PLA-BaG composites. They chose a silicate and phosphate BaG to determine the influence of the BaG composition on the overall degradation behavior of the composites.

The topic is of great interest, the investigations are conducted thoroughly and extensively, ranging from mechanical properties, to degradation behavior of the polymer as well as ion release from the glasses and visual characterization during and after degradation by SEM/EDX and µCT.

The manuscript is well written, the results are properly presented and discussed. This manuscript can be published as is.

Thank you for your comments.

Reviewer 3 Report

The manuscript entitled “Impact of Glass Composition on Hydrolytic Degradation of Polylactide/bioactive Glass Composites” submitted by J. Massera et al. is focused in the hydrolytic degradation of polylactide/bioactive glass composites. The degradation was analysed by GPC, mechanical behaviour, SEM imaging and X-ray micro-computed tomography.

Although the manuscript might seem interesting, my main concern about this work is the novelty. A previous paper from some of the authors (referred also in the manuscript) include similar experiments with the same polymer matrix and Si-BaG glasses with quite close composition. (Niemelä, T.; Niiranen, H.; Kellomäki, M. Self-reinforced composites of bioabsorbable polymer and bioactive glass with different bioactive glass contents. Part II: In vitro degradation. Acta Biomater. 2008, 4, 156–164, doi:10.1016/j.actbio.2007.06.007). What is the novelty of this study with respect to the previously mentioned? This point should be clearly explained to define the scope and importance of the manuscript, which is currently weak.

Other points that should be considered:

Point 1. Pg 3, section 2.1. The authors indicate: ‘The Si-BaG was melted at 1425 °C for 3 hours and the P-BaG at 1100 °C for 30 min. The melt was poured into a graphite mold and annealed at Tg + 15 °C for 30 min for both Si-BaG and P-BaG.’. Please, include the specific value of the temperature used for the annealing. How was the Tg obtained?

Point 2. The processing parameters reported in Table 2 indicate that both the temperature and pressure are different also within a series of composites (P-BaGx composites have been processed with quite different conditions). Why? This point should be explained.

Point 3. Pg. 3, section 2.3. The authors say: ‘The pH was measured only at the end of hydrolysis for the short follow-up periods (24, 48, 72 hours, 1 and 2 weeks) and every two weeks before refreshing the solution at the longer follow-up periods (4, 6, 8, 10, 20, 30 and 40 weeks).’ What is the meaning of ‘at the end of hydrolysis’?. Please, explain what you mean, it is confusing.

Point 4. pg. 4, section 2.3.5. Indicate the dimensions of the samples used to analyse the mechanical properties.

Point 5. Pg. 5, section 3.1. (second paragraph). The authors indicate that there is a large deviation in glass content in P-BaG composites. The explanation of this fact seems speculative. Some references should be included to support it.

Point 6. Significant differences should be included in Figures 1 and 2 (at least).

Point 7. Pg. 6, section 3.1. From my point of view, the following sentence suddenly appears virtually out of context. The text should be rewritten to include the sentence in an appropriate context.

‘Melt compounding S53P4 glass with PLA is challenging. Blaker et al. reported and we confirmed in our processing trials that extruding composites of S53P4 glass and PDLLA induced bubble formation, amber color, sweet odor, and over 85% decrease in the polymer molecular weight at a reasonable processing temperature of 150 °C, suggesting extensive thermal degradation.’

Point 8. Pg. 6, section 3.1. Additional references should be included to support the explanation. Otherwise, it seems very speculative:

‘This high drop in molecular weight was not anticipated. However, it may partly be due to the different extruder configuration compared to the processing set-up in [21] and the slight plasticizing effect of the Si-BaG on the polymer melt. The possible reasons for the drop in the Mw of PLA70 could be high intrinsic water content in some bioactive glasses or partial glass dissolution in the polymer melt. It appears that with the Si-BaGs, the more reactive the glass is, the more readily the excessive thermal degradation of the polymer occurs.’

Point 9. The water absorption of P-BaG composites remains constant after 20 weeks (Figure 4), however the release of ions (Figure 6) does no show significant changes at this point. Why there is no correlation between water absorption results and ion release?

Some other minor points:

  1. Please check the second sentence of the abstract. There is a typing error.

Author Response

Reviewer 3

The manuscript entitled “Impact of Glass Composition on Hydrolytic Degradation of Polylactide/bioactive Glass Composites” submitted by J. Massera et al. is focused in the hydrolytic degradation of polylactide/bioactive glass composites. The degradation was analysed by GPC, mechanical behaviour, SEM imaging and X-ray micro-computed tomography.

Although the manuscript might seem interesting, my main concern about this work is the novelty. A previous paper from some of the authors (referred also in the manuscript) include similar experiments with the same polymer matrix and Si-BaG glasses with quite close composition. (Niemelä, T.; Niiranen, H.; Kellomäki, M. Self-reinforced composites of bioabsorbable polymer and bioactive glass with different bioactive glass contents. Part II: In vitro degradation. Acta Biomater. 2008, 4, 156–164, doi:10.1016/j.actbio.2007.06.007). What is the novelty of this study with respect to the previously mentioned? This point should be clearly explained to define the scope and importance of the manuscript, which is currently weak.

The novelty of the research is comparing BaGs with different compositions (Si-BaG and P-BaG) on the degradation characteristics of the composites. While much work has been performed using silicate bioactive glasses, little is known when phosphate glasses are used in composites. This is of outmost importance since the dissolution mechanism of the phosphate glasses differ from the typical silicate ones. The introduction was modified to clarify the novelty of the study.

Other points that should be considered:

Point 1. Pg 3, section 2.1. The authors indicate: ‘The Si-BaG was melted at 1425 °C for 3 hours and the P-BaG at 1100 °C for 30 min. The melt was poured into a graphite mold and annealed at Tg + 15 °C for 30 min for both Si-BaG and P-BaG.’. Please, include the specific value of the temperature used for the annealing. How was the Tg obtained?

The specific annealing temperatures and the method for obtaining the Tg were added.

Point 2. The processing parameters reported in Table 2 indicate that both the temperature and pressure are different also within a series of composites (P-BaGx composites have been processed with quite different conditions). Why? This point should be explained.

The Si-BaG had a plasticizing effect on the polymer melt, as mentioned on p. 6. This plasticizing effect led to decreased pressure in the extruder, which was compensated by lowering the processing temperatures. The different glass contents had different effects on the viscosity of the polymer melt.

A few sentences has been added in the material and methods for clarification.

Point 3. Pg. 3, section 2.3. The authors say: ‘The pH was measured only at the end of hydrolysis for the short follow-up periods (24, 48, 72 hours, 1 and 2 weeks) and every two weeks before refreshing the solution at the longer follow-up periods (4, 6, 8, 10, 20, 30 and 40 weeks).’ What is the meaning of ‘at the end of hydrolysis’?. Please, explain what you mean, it is confusing.

The text was clarified.

Point 4. pg. 4, section 2.3.5. Indicate the dimensions of the samples used to analyse the mechanical properties.

The mechanical properties were analysed from the hydrolyzed samples, so the rods’ dimensions were 2 x 70 mm. The dimensions were added to the mechanical properties – section.

Point 5. Pg. 5, section 3.1. (second paragraph). The authors indicate that there is a large deviation in glass content in P-BaG composites. The explanation of this fact seems speculative. Some references should be included to support it.

A reference has been added in regard to phosphate glasses being prone to hydrolysis by environmental moisture. Unfortunately, only little work has been done on the co-extrusion of PLA and phosphate glasses. Also, most of the work have used smaller particles of composition much more stable to moisture that the one we have used. Finally, in general authors only report either the targeted glass content

https://journals.sagepub.com/doi/full/10.1177/0892705717729014

Or the measured glass content

https://www.sciencedirect.com/science/article/pii/S1742706109003584, which brings difficulties for comparing the values.

Point 6. Significant differences should be included in Figures 1 and 2 (at least).

Statistical analysis was not possible to conduct because of the small sample size.

Point 7. Pg. 6, section 3.1. From my point of view, the following sentence suddenly appears virtually out of context. The text should be rewritten to include the sentence in an appropriate context.

‘Melt compounding S53P4 glass with PLA is challenging. Blaker et al. reported and we confirmed in our processing trials that extruding composites of S53P4 glass and PDLLA induced bubble formation, amber color, sweet odor, and over 85% decrease in the polymer molecular weight at a reasonable processing temperature of 150 °C, suggesting extensive thermal degradation.’

The text was modified to make it better flowing: “Melt compounding PLA with BaGs is challenging and often leads to extensive thermal degradation of the polymer. Blaker et al. reported and we confirmed in our processing trials that…”

Point 8. Pg. 6, section 3.1. Additional references should be included to support the explanation. Otherwise, it seems very speculative:

‘This high drop in molecular weight was not anticipated. However, it may partly be due to the different extruder configuration compared to the processing set-up in [21] and the slight plasticizing effect of the Si-BaG on the polymer melt. The possible reasons for the drop in the Mw of PLA70 could be high intrinsic water content in some bioactive glasses or partial glass dissolution in the polymer melt. It appears that with the Si-BaGs, the more reactive the glass is, the more readily the excessive thermal degradation of the polymer occurs.’

A reference was added.

Point 9. The water absorption of P-BaG composites remains constant after 20 weeks (Figure 4), however the release of ions (Figure 6) does no show significant changes at this point. Why there is no correlation between water absorption results and ion release?

The release of the ions, from the glass to the medium, is delayed due to the polymer which acts as a barrier for the ion diffusion. The text has been modified in the discussion to further clarify this.

Some other minor points:

  1. Please check the second sentence of the abstract. There is a typing error.

The authors could not locate the typing error.

Round 2

Reviewer 3 Report

In the revised manuscript, the authors have addressed my comments and suggestions.

Author Response

Thank you to the reviewer 3 for all the valuable comments he provided in the first round and we are happy to see him satisfy.

Thank you and best regards on behlaf of all co-authors.